# Effects of Immobilization and Re-Mobilization on Knee Joint Arthrokinematic Motion Quality

**DOI:** 10.3390/jcm9020451

**Published:** 2020-02-06

**Authors:** Dawid Bączkowicz, Grzegorz Skiba, Krzysztof Falkowski, Przemysław Domaszewski, Noelle Selkow

**Affiliations:** 1Faculty of Physical Education and Physiotherapy, Opole University of Technology, 45-007 Opole, Poland; 2Opole Rehabilitation Center, 48-317 Korfantów, Poland; 3Department of Orthopaedics, University Clinical Hospital in Opole, 45-401 Opole, Poland; 4School of Kinesiology and Recreation, Illinois State University, Normal, IL 5120, USA

**Keywords:** immobilization, re-mobilization, arthrokinematics, crepitus, synovial joint, cartilage

## Abstract

Background: Knee immobilization is a common intervention for patients with traumatic injuries. However, it usually leads to biomechanical/morphological disturbances of articular tissues. These changes may contribute to declining kinetic friction-related quality of arthrokinematics; however, this phenomenon has not been analyzed in vivo and remains unrecognized. Thus, the aim of the present study is to investigate the effect of immobilization and subsequent re-mobilization on the quality of arthrokinematics within the patellofemoral joint, analyzed by vibroarthrography (VAG). Methods: Thirty-four patients after 6-weeks of knee immobilization and 37 controls were analyzed. The (VAG) signals were collected during knee flexion/extension using an accelerometer. Patients were tested on the first and last day of the 2-week rehabilitation program. Results: Immobilized knees were characterized by significantly higher values of all VAG parameters when compared to controls (*p* < 0.001) on the first day. After 2 weeks, the participants in the rehabilitation program that had immobilized knees showed significant improvement in all measurements compared to the baseline condition, *p* < 0.05. However, patients did not return to normal VAG parameters compared to controls. Conclusion: Immobilization-related changes within the knee cause impairments of arthrokinematic function reflected in VAG signal patterns. The alterations in joint motion after 6 weeks of immobilization may be partially reversible; however, the 2-week physiotherapy program is not sufficient for full recovery.

## 1. Introduction

Immobilization is a common orthopedic intervention for patients with severe traumatic injuries, and despite the benefit, immobilization usually leads to dysfunction in articular and extra-articular tissues of diarthrodial (synovial) joints [1,2]. One of these dysfunctions is joint contracture, which is characterized by loss of passive range of motion (ROM), and is a common clinical problem in orthopedics and rehabilitation medicine [3,4]. In the development of joint contractures that result from long-term immobilization, shortening of the joint capsule, synovial adhesions and arthrofibrosis play decisive roles, and may present as a generalized joint stiffness [5,6]. On a cellular level, arthrofibrosis is characterised by upregulated myofibroblast proliferation with reduced apoptosis, adhesions, aggressive synthesis of extracellular matrix that can fill and contract joint pouches and tissues, and often also heterotrophic ossification [7]. Moreover, joint immobilization and unloading result in chondral softening, proteoglycan loss and overall reduction of the cartilage thickness [8,9]. Therefore, it seems that immobilization-related biomechanical and morphological disturbances not only lead to deterioration in ROM but also contribute to declining quality of arthrokinematic motion, which refers to the palpatory sense of how smoothly a joint can be moved through its ROM [10,11,12]. This hypothesis supports clinical observations of many patients reporting popping, snapping or grinding sensations, especially within the anterior knee compartment [6,10,13,14,15,16,17].

In contrast to quantitative assessment of joint motion via ROM measurements (using a goniometer or arthrometer), the analysis of arthrokinematic motion quality is highly subjective in nature [18]. To date, this physical evaluation has focused on palpation analysis of the integrity and smoothness of motion regarding the presence or absence of crepitations (mechanical oscillations and sounds produced by articular surfaces) [19]. Therefore, there have been calls for the development of more specific methods to evaluate the quality of arthrokinematic motion [20].

One of the novel methods developed for more sensitive and objective assessment of articular function related to crepitus occurrence is vibroarthrography (VAG) [21,22]. This noninvasive method could be a helpful tool for clinicians, especially for rehabilitation specialists who are concerned with the analysis of arthrokinematic motion quality [10]. The VAG method is based on the analysis of high frequency vibroacoustic emission, which is a natural phenomenon acquired from relative motion of articular surfaces [16]. Although the VAG method is still in development, it shows high accuracy, sensitivity and specificity [23,24,25,26]. Recently it has been used to differentiate disorders of the patellofemoral joint (PFJ), due to the specific, disorder-related character of the VAG signal pattern [16,17]. It was also suggested that the analysis of arthrokinematic motion quality may be a helpful tool to monitor the effectiveness of specific rehabilitation in joint disorders [10].

Therefore, the purpose of this study was to evaluate the arthrokinematic motion quality of the knee in patients after knee joint immobilization via vibroarthrography. Moreover, we investigated the influence of a rehabilitation program for restoration of the proper quality of joint movement in the knee joint. The findings could broaden the knowledge about immobilization-related restrictions in knee arthrokinematics and contribute to improvement in medical care, especially in utilizing appropriate techniques of physical therapy for patients after knee immobilization.

## 2. Method

### 2.1. Study Population

The study group included thirty-four patients between the ages of 30 and 40 years old after 6 weeks of knee joint immobilization. Individuals were treated non-operatively (13 cases) or operatively (21 cases) following a simple tibial shaft fracture, diagnosed as type 42-A by the AO/OTA classification of tibial diaphyseal fractures [27]. Individuals with a prior history of knee injury, complications of fracture healing or diagnosed disorders and posttraumatic syndromes within the knees were excluded from the study, to prevent any signal artifact from disorders not associated with knee immobilization. Moreover, due to the methodology of the VAG assessment, only cases with the range of knee motion from 0 of extension to 100 degrees of flexion in the injured lower limb were enrolled in the study. Finally, the analyzed group included 34 patients, who were tested twice: on the first and last day of the rehabilitation program, which started 1 week after the cast removal.

The control group consisted of 37 healthy subjects, who were tested once. There were no statistically significant differences in anthropometric indices in comparison to the group of immobilized patients. By self-report, control subjects had neither a history of lower extremity injury nor other pathology within the knee joint. For detailed characteristics of all participants, see Table 1.

All patients were treated as inpatient or outpatient in Opole Rehabilitation Centre. Signed informed consent was obtained from all patients, and the rights of subjects were protected. The project was approved by the Opole Voivodship Ethics Committee (ethical approval code No. 202/06.06.2013).

### 2.2. Medical Procedures and Rehabilitation Program

In accordance with the medical procedures, after closed or open (with internal fixation) reduction, all patients received a long-leg cylinder plaster cast, in which the injured limb was immobilized for 6 weeks. The knee joints were immobilized in 10–15 degrees of flexion and the ankle joint in the neutral position. After a satisfactory result of the control medical examination and after a week of cast removal, all patients underwent an outpatient, individual functional rehabilitation program conducted by the physiotherapists, in a single rehabilitation center. The rehabilitation program was continued for two weeks, and in total, there were 10 sessions, two hours each. The physiotherapy was focused on lower-limb muscle-strengthening exercises, range of motion restoration and proprioceptive training (a detailed rehabilitation program is presented in Appendix A).

In addition, at the beginning and at the end of the rehabilitation program, the ROM of the knee joint was tested. Passive knee flexion and extension was measured with the patients lying in the supine position. Knee ROM was measured using a standard goniometer with the axis placed over the lateral epicondyle of the femur, the proximal arm aligned with the greater trochanter of the femur, and the distal arm aligned with the lateral malleolus of the ankle.

### 2.3. Assessment of Arthrokinematic Motion Quality

Performed analyses were based on standardized methodology described previously [10,11,16,17]. Briefly, for each knee, assessment of arthrokinematic motion quality was performed in an open kinetic chain in flexion/extension motion using an acceleration sensor placed 1 cm above the apex of the patella. In a sitting position, the following procedure was performed: (i) loose hanging legs with knees flexed at 90°; (ii) full knee extension from 90° to 0°; (iii) re-flexion (from 0° to 90°), four times in a 6-second period. The constant velocities of both extension–flexion motion and measuring condition were maintained at 82 beats per minute with a metronome. The angle of the knee joint was measured using an electro-goniometer, but because the VAG signal might be distorted by the electro-goniometer placement, which could generate noise signal, this procedure was only used during determination of the experimental condition before relevant tests. In the immobilized and control groups both limbs were tested in a random order.

The VAG signals generated were collected using an acceleration sensor, model 4513B-002, with a multi-channel Nexus conditioning amplifier (Brüel & Kjær Sound & Vibration Measurement A/S, Nærum, Denmark). The signals were recorded as a time series expressed in volts, with the frequency range of 0.7–1000 Hz and a sampling rate of 10 kHz. Each result was high-pass filtered according to the 50 Hz threshold, to minimize artifacts, e.g., muscle tremor. The obtained signals ranged from 50 to 1000 Hz and were described using four parameters. The variability of the VAG signal was assessed by computing the mean-squared values of the obtained signal in fixed-duration segments of 5 ms each, and then computing the variance of the values of the parameter over the entire duration of the signal (VMS parameter) [16,28]. Moreover, for signal amplitude analysis, the R4 parameter was used. Because of the four full flexion/extension motion cycles, the R4 parameter was calculated as the difference between the mean of four maximal values and the mean of four minimal VAG signal values [17].

The frequency characteristics of the VAG signal were examined by short-time Fourier transform analysis. The short-time spectra were obtained by computing the discrete Fourier transform of segments with 150 samples each, the Hanning window and the 100 samples overlap of each segment. The spectral activity was analyzed by summing the spectral power of the VAG signal in two bands: 50–250 Hz (P1 parameter) and 250–450 Hz (P2 parameter) [10,11].

For a better understanding of how the distinct features extracted from the signal provide the particular information about the signal variability and its range, representative vibroarthrographic signals are presented in Figure 1. It showcases plots of preprocessed signals (reflecting recorded vibrations expressed in volts) with visually-defined VMS and R4 parameters. In turn, for the convenience of the signal frequency domain analysis, respective spectrograms (visual representation of the spectrum of frequencies of a signal as it varies with time) are presented, with the selected P1 and P2 parameter bands (Figure 2).

### 2.4. Statistical Analysis

The normality test was performed using the method of Shapiro–Wilk. Demographic characteristics of analyzed groups are presented as descriptive statistics (mean and standard deviation), and the Mann–Whitney U test was used to evaluate differences between them. Differences in VAG and ROM dependent variables after logarithmic transformation (due to the not-normally-distributed data) were analyzed with the one-way analysis of variance. When significant interactions were identified, Tukey range analyses for unequal sample size and Tukey honest significant difference were applied as post-hoc tests. For the correlation between VAG parameters and knee range of motion, the Spearman’s rank correlation coefficient test was performed. *p*-values ≤ 0.05 were considered as significant. Statistics were analyzed using Statistica v.12 (StatSoft, Inc., Tulsa, OK, USA).

## 3. Results

The mean values of the VAG signal parameters registered from 34 patients and 37 control subjects are presented in Table 2. Immobilized knees, when analyzed before rehabilitation, generate signals characterized by significantly higher values of amplitude parameters (VMS and R4), both in comparison to the contralateral, non-immobilized side and to the knees of the control group. It is also seen on the representative plots of the VAG signals, where the knees after immobilization were characterized by higher amplitude and variability of the signal (which indicates a higher vibration level) as compared to healthy subjects (Figure 1). Analogous, time-frequency analysis of the representative spectrograms showed substantial differences between immobilized and control signals (Figure 2). In this example, it can be observed that immobilized knees before rehabilitation generated a signal with higher power, especially in the frequency range of 50–250 Hz and 250–450 Hz. Respectively, this corresponds with two-fold higher values of P1 and P2 parameters, when compared to control subjects and contralateral, non-immobilized knees (Table 2). In contrast to the immobilized group, no difference between sides was observed within healthy controls.

After 2 weeks, the participants of the rehabilitation program had significant improvements in all VAG measurements within the immobilized knees, as compared to day 1 of rehabilitation (Table 2). It is also apparent in Figure 1b and Figure 2b, where the variability and range of the sample signal and its spectral power were lower, which corresponds with a lower vibration level. However, the values of R4 and P2 parameters were not equal with the values obtained from control knees (both from the control group and contralateral, non-immobilized knees) (Table 2). The rehabilitation program had no effect on the values of VAG parameters analyzed for the non-immobilized knee joint in the immobilized group.

After immobilization, mean knee flexion in the immobilized leg was significantly reduced compared with the non-immobilized side (115.7° ± 13.3 vs. 142.6° ± 4.3, respectively, *p* < 0.001). On the other hand, the period of immobilization did not affect knee extension. After the rehabilitation program, the range of knee flexion significantly increased by 15.6° ± 10.3 (*p* < 0.05), when compared to the baseline assessment. In addition, interactions between arthrokinematic motion quality and knee ROM, collected before and after the rehabilitation program, were analyzed. There was a significant negative correlation between amplitude VAG parameters (VMS and R4) and the range of knee flexion, but only in the analysis before rehabilitation (Table 3).

## 4. Discussion

The purpose of this study was to examine the effect of knee immobilization and the subsequent rehabilitation program on arthrokinematic motion quality. Additionally, we analyzed the relationships between values of VAG parameters and knee ROM. It is difficult to compare our results to previous studies, as to the best of our knowledge, no prior research has investigated the immobilization-related deteriorations of qualitative aspects of motion within human joints. We have previously studied the impact of particular disorders of the PFJ (chondromalacia, osteoarthritis and patellar compression syndrome) on arthrokinematic motion quality. The results showed that VAG signals generated by disordered knees significantly differ from healthy knees. We postulated that the mentioned observation may be a result of biomechanical disturbances including cartilage deterioration, chondral stress and patellar maltracking, respectively, with the biomechanical and morphological background of the mentioned diseases.

The results presented herein show that the 6-week period of immobilization negatively affects the quality of arthrokinematic motion analyzed by the VAG method. Signals registered from knees after immobilization possessed greater variability, amplitude and higher spectral power. In contrast, the signals generated by the control joints were relatively smooth and have a lower power at lower and higher frequencies. This observation is expressed as higher values of VAG signal parameters in immobilized joints than in control knees. Therefore, we assume that the above-mentioned diminished arthrokinematic motion quality is associated with several biomechanical impairments, postulated as complications of joint immobilization. It has been found in numerous animal models that 15 days of immobilization caused retrogressive alterations of articular cartilage, while 8 weeks of immobilization induced significant changes in cartilage thickness, number of chondrocytes, extracellular matrix components, and cartilage surface irregularity, fibrillation and softening [29,30,31,32]. Furthermore, immobilization may result in limited production (and degraded quality) of synovial fluid and hinder its diffusion in the joint cavity, which negatively influences lubrication of the articular surfaces [33,34]. As a consequence, an increase in the coefficient of friction occurs during patello-femoral motion, observed as increased vibrations registered in the VAG signal.

Moreover, it must be taken into consideration that the immobilized patients had limitations in knee flexion. Experimental research has shown that two factors are at play in the development of immobilization-induced joint contractures: arthrogenic (bone, cartilage, synovial membrane, capsule, and ligaments) and myogenic [35]. It was demonstrated that among arthrogenic components, shortening of the joint capsule plays an important role in the development of ROM restrictions [36,37]. Therefore, it is possible that knee capsular contracture caused an increase of patellofemoral stress, which contributed to the increase of friction [38]. This assumption may explain the presence of negative correlations between knee flexion ROM and values of VAG amplitude parameters (VMS and R4). However, this should be confirmed by further studies, since all the relations between biomechanics of noncontractile tissues, compressive stress of articular surfaces and kinetic friction are not known. Nonetheless, we showed that the presence of immobilization-related capsular contracture may limit both ROM of the knee and arthrokinematic motion quality.

The post-rehabilitation condition in comparison to the baseline assessment was characterized by significantly lower values of all analyzed VAG parameters, which reflects an improvement in arthrokinematic motion quality. These results agree with findings obtained in animal models, indicating that the combination of therapeutic exercises aided the restoration of ROM, improved the flow of synovial fluid in the joint cavity, and aided the nutrition of the cartilage, providing regeneration for articular structures [34,37]. It was shown that re-mobilization promotes recovery of morphological organization and improvement of morphological parameters of hyaline cartilage [30,39]. Nonetheless, despite completion of the 2-week rehabilitation program focused on improving knee function, the re-mobilized knees had higher values of R4 and P2 parameters than the control knees. This demonstrates that related signals possess greater amplitude and higher spectral power in the range of 250–450 Hz, which corresponds to the deteriorated arthrokinematic motion quality. This observation may be confirmed by the study of Iqbal et al. [40], who analyzed the morphological changes in the cells of articular cartilage of the patella on immobilized and re-mobilized rat knees. They found, that after 8 weeks of re-mobilization after four weeks of immobilization, reversible changes were observed in the superficial zone of articular cartilage; however, regenerative processes were still active. Thus, it seems that the 2-week rehabilitation program in this study is insufficient to reverse changes of immobilization on synovial joint function.

The study limitation includes lack of long-term follow-up data (only a 2-week rehabilitation program) and a heterogenous group including operative and non-operative patients. Although all subjects analyzed in this study underwent an X-ray examination, the direct status of articular structures was not assessed by a suitable imaging method, e.g., magnetic resonance imaging, and remains unknown [41]. Therefore, subsequent research of arthrokinematic motion quality should include evaluation of the condition of, for example, hyaline cartilage to establish the real relationship between immobilization-related chondral disturbances and VAG signals. Moreover, due to the specific inclusion criteria, the clinical treatment of patients is not the typical treatment utilized for this injury. It should also be considered, that our study did not include a group of immobilized patients who were not undergoing rehabilitation, which is a limitation in drawing a conclusion about direct impact of an applied rehabilitation program on the arthrokinematic motion quality.

In summary, the presented results extend knowledge on the consequences of knee joint immobilization and its further re-mobilization. We conclude that immobilization-related changes within the knee joint environment cause impairments of arthrokinematic function reflected in VAG signal patterns. In addition, our findings suggest that the alterations induced in joint motion quality by 6 weeks of knee immobilization may be partially reversible; however, the 2-week physiotherapy program is not sufficient for full recovery, and it should be considered that a longer duration of re-mobilization may be required to restore the proper arthrokinematic motion quality.

## Figures and Tables

**Figure 1 jcm-09-00451-f001:**
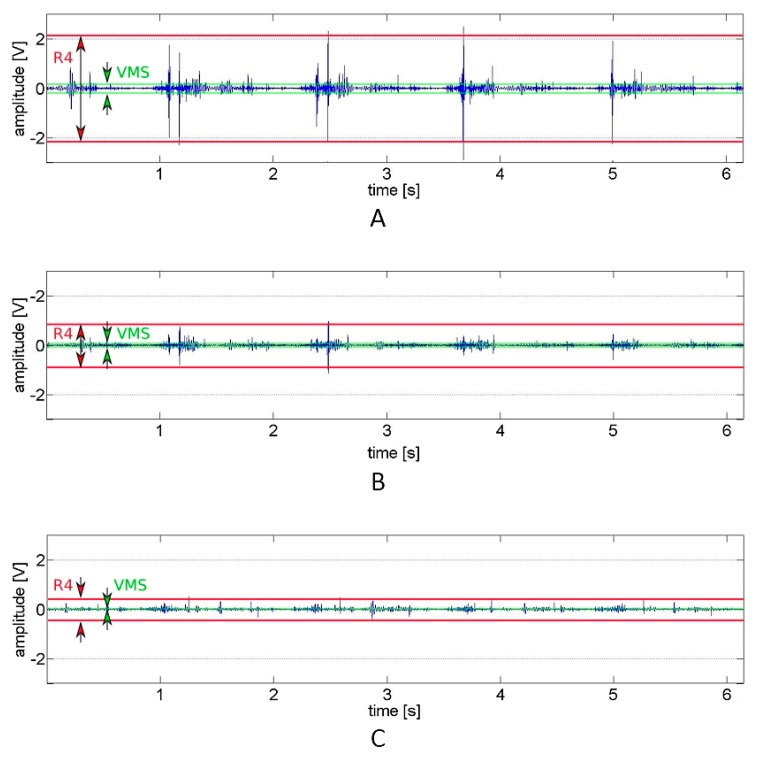
The course of representative vibroarthrographic signals for (**A**) knees after immobilization, (**B**) knees after rehabilitation (**C**) control healthy knees. This showcases plots of preprocessed signals, reflecting recorded vibrations expressed in volts with visually defined VMS (variance of the mean squares) and R4 (mean of 4 maximal and 4 minimal values) parameters. Higher signal amplitude means more prominent vibrations.

**Figure 2 jcm-09-00451-f002:**
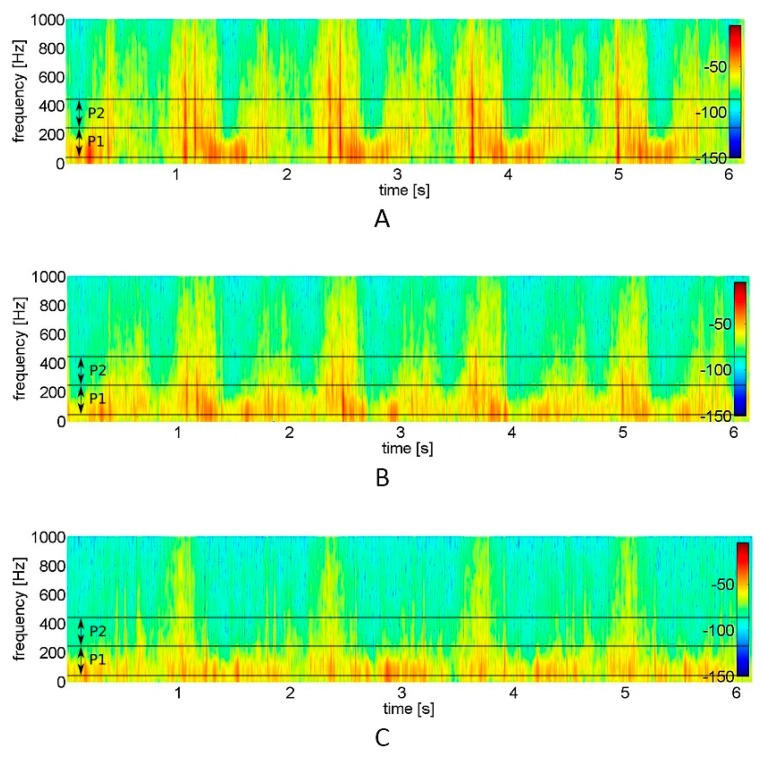
Spectrograms of the vibroarthrographic signals representative for (**A**) knees after immobilization, (**B**) knees after rehabilitation (**C**) control healthy knees. Short-time Fourier transform analysis serves visual representation of the spectrum of frequencies of a signal as it varies with time. Analyzed frequency bands are marked with lines (P1, power spectral density band of 50 to 250 Hz; P2, power spectral density band of 250 to 450 Hz).

**Table 1 jcm-09-00451-t001:** Characteristics of the study groups.

	Immobilized Group (*n* = 34)	Controls (*n* = 37)
Number of males/females	23/11	24/13
Age (years), mean (SD)	33.8 (2.7)	34.2 (2.9)
Height (cm), mean (SD)	176.8 (10.5)	175.4 (11.2)
Weight (kg), mean (SD)	75.4 (12.1)	76.2 (13.8)
BMI, mean (SD)	24.6 (3.4)	25.3 (3.7)
Open/closed fracture	3/31	-
Tscherne classification for open fractures		-
*Grade I*	1	
*Grade II*	2	
Tscherne classification for closed fractures		-
*Grade 0*	11	
*Grade I*	17	
*Grade II*	3	
Fibula fracture, no/yes	9/25	-
Managed operatively/non-operatively	21/13	-

Abbreviations: BMI, body mass index.

**Table 2 jcm-09-00451-t002:** Values of VAG parameters in patient and control groups.

	VMS (V) Mean (SD)		R4 (V) Mean (SD)		P1 (V^2^/Hz) Mean (SD)		P2 (V^2^/Hz) Mean (SD)	
Controls	L	R	L	R	L	R	L	R
	0.072 (0.213)	0.075 (0.194)	2.50 (2.30)	2.58 (2.48)	6.12 (9.73)	6.22 (11.40)	0.62 (1.01)	0.66 (1.52)
Patients	IMB	non-IMB	IMB	non-IMB	IMB	non-IMB	IMB	non-IMB
*before rehabilitation*	0.137 (0.401) *	0.081 (0.243)	3.98 (3.57) *	2.61 (2.86)	11.89 (15.91) *	6.30 (11.71)	1.36 (2.32) *	0.71 (1.71)
*after rehabilitation*	0.082 (0.219)	0.076 (0.212)	3.01 (2.99) *	2.53 (2.28)	7.75 (12.84)	6.39 (11.18)	0.87 (1.50)	0.70 (1.65)
*p*-value								
*controls vs. before rehabilitation*	**<0.001**	0.18	**<0.001**	0.52	**<0.001**	0.34	**<0.001**	0.49
*controls vs. after rehabilitation*	0.11	0.55	**0.047**	0.44	0.09	0.61	**0.045**	0.46
*before rehabilitation vs. after rehabilitation*	**0.026**	0.18	**0.008**	0.57	**0.016**	0.58	**0.032**	0.72

Abbreviations: *, difference statistically significant in comparison to non-immobilized knees; L, R, left and right knees of controls, respectively; IMB, non-IMB, immobilized and non-immobilized knees of patients, respectively; VMS, variance of the mean squares calculated in 5 ms windows; R4, the difference between the mean of four maximum and the mean of four minimum values; P1, P2, power spectral density bands: 50–250 Hz and 250–450 Hz, respectively.

**Table 3 jcm-09-00451-t003:** Correlations between range of knee motion and VAG parameters in the immobilized group.

			VAG Parameters			
			VMS	R4	P1	P2
Range of motion	before rehabilitation	r	**−0.458**	**−0.520**	−0.431	−0.382
		p	0.036	0.003	0.07	0.05
	after rehabilitation	r	−0.153	−0.316	−0.084	−0.046
		p	0.11	0.08	0.28	0.37

Abbreviations: VMS, variance of the mean squares calculated in 5 ms windows; R4, the difference between the mean of four maximum and the mean of four minimum values; P1, P2, power spectral density bands: 50–250 Hz and 250–450 Hz, respectively.

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
