# Peer review of "Effects of Immobilization and Re-Mobilization on Knee Joint Arthrokinematic Motion Quality"

_jcm, 2020, doi:10.3390/jcm9020451_

Round 1

Reviewer 1 Report

The authors addressed my concerns. 

Author Response

Thank you.

Reviewer 2 Report

General Summary:

The authors looking at the kinematics of the knee joint after a period of prolonged immobility in a long leg cast for 6-weeks after the non-operative or operative treatment of type 42-A tibial shaft fractures or simple tibial shaft fractures and again after a 2-week rehabilitation protocol. Authors utilized the vibroarthrography to determine the ability of the knee joint to move through range of motion, and due to the significant contribution of the patellofemoral joint to the mobility of the knee joint through flexion and extension, extrapolated that this demonstrates the ability of the patellofemoral joint function after prolonged immobilization. This work represents objective data of current clinical phenomenon, that those with prolonged immobilization demonstrate altered biomechanics that live on a spectrum from normal kinematics to joint contractions and frank arthrofibrosis, that are evident however difficult to quantify and demonstrate. This work represents a reproducible measurement tool and the basis for future studies to look at joint kinematics in a myriad of different orthopedic conditions including the effects of immobilization after fracture surrounding several joints.

Broad Comments:

This paper demonstrates one of the first utilizations of a very clinically useful measure, vibroarthrography, after prolonged immobilization and assists in the validation of the measure for future use. However, the clinical applicability is limited as the clinical phenomenon of arthrofibrosis and joint contracture after immobilization is while known which is why the large majority of simple tibial shaft fractures are treated operatively with intra-medullary nails which allows for immediate weight-bearing or, at the very least, passive range-of-motion. This brings into question the clinical treatment of patients in this study as they are not the typical treatment utilized for this injury. In addition, the rehabilitation protocol published with this study is a gentle return to mobility lasting for only 2-weeks, which is not the typically rehabilitation protocol clinically utilized in this patient population. Lastly, the extrapolation that the joint kinematics most represents the movement through the patellofemoral is an over-simplification. Those who do arthroscopic surgery to remove arthrofibrosis after large trauma and extended immobilization have documented that is not only the mobility of the patellofemoral joint but the ability of the capsule, posterolateral corner and ability of the tibia to rotate to accommodate the femoral condyles which affects and restricts extension-flexion of the knee joint. As such, the clinically implication insinuated in this paper that, despite a full course of intensive rehabilitation, patients may never return to completely normal joint kinematics within the patellofemoral joint is extremely misleading.

A review on each section in the manuscript is found below. 
Specific Comments:

TITLE                                                    
The title is clear and demonstrates the research question and study objective clearly.   
ABSTRACT
The abstract is accurate, concise and specific, coherent and readable, and structured. 

- There should be mention within the abstract that patients do not return to normal VAG paraments compared to controls.
INTRODUCTION

Previous pertinent literature is cited and discussed with regards to vibroarthrography, the purpose/research hypotheses is clearly stated, and the conceptualization and rationale of the study is clearly apparent. However, the extrapolation that the PFJ joint is the most clinically effected or the primary dysfunction with immobilization is weak and should be removed. A better literature search and description of arthrofibrosis after immobilization should be completed ad described.
- Line 44-45: There references do discuss the effect of the patellofemoral joint on kinematics however are not specific to the effect of immobilization on the patellofemoral joint and cannot be used to extrapolate that the phenomenon of joint contracture/arthrofibrosis preferentially afters the PFJ.
- Line 47-48: This statement needs a reference. Clinical experience demonstrates that this is arthrofibrosis that develops between the patella and femur which limits contact of the patella to the femur.
METHODS     
The study design is appropriate to achieve the study objective and the study population is clearly and adequately described. The sampling procedures are sufficiently described, and the statistical analyses is appropriate and used appropriately. The statistical analysis is performed accurately. However, there is significant selection bias and the patient population limits the clinical applicability and utility of this paper. Those operatively treatment with intramedullary nails are typically allowed to mobilize immediately or at least allow passive/active range-of-motion, this is the benefit of IM nails as opposed to non-operative treatment and may be the main reason patients clinically improve with IM nailing versus non-operative treatment. This brings into question the clinical treatment of these patients. In addition, rehabilitation was started within a week of cast removal instead of immediate which is strikingly different to clinical practice which supports rehabilitation as soon as clinically safe. In addition, the rehabilitation utilized is short, only 2 weeks, and is very gentle return to range of motion. This rehabilitation protocol is far from optimized and may be the pain reason these patients do not return to baseline kinematics.
- Table 1: This should include additional clinical information including soft-tissue damage or Tscherne scale considering some were open fractures, mechanism of injury, additional injuries including head injuries and etc., Union rates and number of mal-unions or non-unions, number that were operatively managed versus non-operatively managed.    
RESULTS                    
The results are clearly presented, the statistics are tested adequately, and the text does not duplicate tables. 
DISCUSSION
Support of the hypothesis is noted, previous pertinent literature is critiqued, and the continuity between the present study and previous work is demonstrated. The similarities and differences to other studies have been noted and limitations of the study have been noted.
However there needs improvement of the discussion and conclusion regarding the contribution of the work and avenues for future research. More specifically, the final concluding paragraph does not reference the reliability of digital algometry and its contribution or avenues for future research however focuses on secondary outcomes of differences in digital algometry between healthy piriformis muscles and non-healthy piriformis muscles. 
Line 237-238: “Therefore, it is possible that knee capsular contracture caused an increase of patellofemoral stress, which contributed to the increase of friction within the PFJ.” – There is no reference to this statement, and it can be argued that

Line 259-261: “Therefore, physicians should be aware that lower extremity immobilization, may show long-term deficits in the form of higher kinetic friction within synovial joints and as a result have lower resistance to load transfer, despite the lack of primary joint injury.” – This statement must be removed. The use of long-term deficits implies that there is prolonged or near-permanent effect of immobilization despite adequate treatment, which a gentle 2-week rehabilitation program is not.

Line 264 – Paragraph regarding limitations needs to be expanded substantially. There are multiple limitations to the studies at hand, including no long-term follow up, lack of wide-spread clinical use of vibroarthrography, heterogenous group including operative and non-operative patients, etc.

FORM, STYLE, AND SUBSTANCE
The manuscript is clearly written, however is can improve organization of the introduction.

REFERENCES
The references are relevant and comprehensive and in the correct format.
Tables: Table 2 should be moved to be above table 3. Currently coming after table 3 makes this confusing.

Figure 1 and 2: The legends should be altered to describe exactly what is being see in these diagrams. Considering this is in a clinical journal, most clinicians will not be able to independently interpret what these figures are trying to represent.

Author Response

This manuscript is a resubmission of an earlier submission. The following is a list of the peer review reports and author responses from that submission.

Round 1

Reviewer 1 Report

Title

Effects of immobilization and re-mobilization on knee joint arthrokinematic motion quality

Brief summary (one short paragraph)

The investigators seek to distinguish impairments of the patellofemoral joint after 6 weeks of knee immobilization using an accelerometer before and after 2 weeks of rehabilitation compared to normal joints. Vibroarthrographic signals recorded above the patella during flexion extension movements were different from controls and partially reestablished with rehabilitation.

Broad comments highlighting areas of strength and weakness.

Strengths:

Interesting experiment using standardized protocols and tools.

Attention provided to a common condition, namely joint changes after immobility where little scientific data exist.

Enough power to detect an effect.

Weaknesses:

The physiological significance of the signal is uncertain. Is it related to changes in cartilage, capsule, bone, muscle weakness, changes in the length tension curve properties of the muscle?

Are changes related to the orthopedic injuries leading to knee immobilization or are they related to the immobilization itself?

Similarly, to which of these tissues should the partial resolution from rehabilitation be attributed?

The tool measures subclinical changes of uncertain clinical significance; patella-femoral impairment after 6 weeks of immobilization is not a common clinical problem.

The investigators correlate accelerometer output from the patellofemoral joint with range of motion from the tibiofemoral joint. The clinical relevance of these correlations is therefore indirect and unclear.

Specific comments

Abstract
30 the investigators do not know that a longer duration of immobilization is required to fully reverse their measures.

The investigation of changes related to knee joint immobility have mainly dealt with the more clinically relevant tibiofemoral joint rather than the patellofemoral joint.

170 the authors refer to figure 1b and figure 2b as evidence of improvement from the rehab program. These figures do not constitute an averaged output from all patients but rather single representative sessions.

202-203 what is meant by this sentence? There was no course of immobilization related signals but rather only one time point after 6 weeks of immobilization.

205-206 what does this sentence means? Which phenomenon?

Numerous sentences in the discussion are awkwardly worded.

206 the hypothesis should be announced in the introduction section.

260 how can the authors conclude that the measured changes are due to increased friction?

Reviewer 2 Report

This was an interesting study that evaluated how immobilization and re-mobilization following knee injury affects the range-of-motion of the joint.  Overall, the study was designed and executed well.  A few questions should be addressed:

Control subjects were reported to not have had any history of lower extremity injury or other pathology.  However, it was not clear if the participants of the study group had any prior injury or history of knee problems that might have complicated the results of this group. Was any analysis done to compare males to females or to correlate results with severity of injury (e.g., open/closed fracture, with/without fibula fracture)?  There appear to be some discrepancies between the explanation of the rehab protocol in the text (lines 98-103) and the details provided in Appendix 1. Specifically, with respect to the details of the at home exercise and the duration of the rehab sessions (i.e., 1 vs 2 hours). It appears to be inferred, but the authors should state that data were not normally distributed, if that was indeed the case. It is a little difficult to understand what the plots in Figures 1 and 2 represent, particularly with those readers who are unfamiliar with the VAG technique.  More information should be provided as well as text to help interpret these plots.  The standard deviations for the output parameters in Table 2 are generally quite large, with some values reaching as high as 2-3 times the mean values.  Can the authors speak to the degree of variability in the data and how that might impact confidence in the group-to-group comparisons? Some of the comparisons in the results are only made between two groups instead of other possible comparisons (e.g., lines 172-3 and lines 182-4); the authors may want to consider adding in other comparisons for full clarity.  The text in lines 241-249 is awkward and difficult to interpret; should be edited.  There are several typos and grammatical mistakes throughout.  One example - in line 306, it should be "3rd" instead of "3th".

Round 2

Reviewer 1 Report

jcm-650403.R1

This manuscript has been returned to the investigators. They provided responses to comments from the reviewers as well as a revised manuscript.

This reviewer is grateful for the opportunity to read a second version of this work. The reviewer takes note of the responses to the various comments.

This reviewer notes that the investigators relate VAG to articular cartilage and synovial fluid arthrokinematic motion and then extended to the "articular environment" to enable specific physical therapy intervention and complement x-ray and MRI.

The investigators also respond that their methods of VAG represents changes of the "whole biomechanical environment of the joint" and that specifically remobilization promotes recovery of "morphological organization and improvement arm of morphological parameters of the hyaline cartilage"

The investigators also correlate accelerometer output of the patellofemoral joint with range of motion of the tibiofemoral joint is justified by "the function of 1 determines the function of the other, and vice versa"

Legends to figures 1 and 2 mislead the reader to think that this is an average signal of multiple knees whereas these are representative signals from one knee at 1 and experimental time.